# PHONATE: Impact of Type-Written Phonological Features of African American Language on Generative Language Modeling Tasks

**Nicholas Deas**
Department of Computer Science
Columbia University
ndeas@cs.columbia.edu

**Jessi Grieser**
Department of Linguistics
University of Michigan
jgrieser@umich.edu

**Xinmeng Hou**
Teachers College
Columbia University
fh2450@tc.columbia.edu

**Shana Kleiner**
School of Social Policy and Practice,
Annenberg School for Communications
University of Pennsylvania
skleiner@upenn.edu

**Tajh Martin**
Department of Computer Science
Columbia University
tm3213@columbia.edu

**Sreya Nandanampati**
Department of Computer Science
Barnard College
sn2875@barnard.edu

**Desmond Patton**
School of Social Policy and Practice,
Annenberg School for Communications
University of Pennsylvania
dupatton@upenn.edu

**Kathleen McKeown**
Department of Computer Science
Columbia University
kathy@cs.columbia.edu

## Abstract

*Warning: This paper contains content and language that may be considered offensive to some readers.* Current Large Language Models perform poorly on African American Language (AAL) texts in tasks like toxicity detection and sentiment analysis. AAL is underrepresented in both pre-training data and existing benchmarks for these tasks, hindering thorough evaluation and understanding of these biases. We introduce a novel approach to synthetically introduce type-written phonological features of AAL into text, a class of AAL features that has been overlooked in prior work. Our goal is to better understand how these features affect generative language models' performance on three tasks: toxicity detection, sentiment analysis, and masked span prediction. We find that finetuning with synthetic type-written phonological features lowers perceived biases on downstream tasks and our ablations reveal which features have particularly large negative impacts on model performance. Our results suggest that phonological features are vital to consider when designing bias mitigation techniques[1].

## 1 Introduction

Existing text resources used in the development and evaluation of Large Language Models (LLMs) lack sufficient representation of linguistic diversity (Dodge et al., 2021). This presents a major obstacle to reliably evaluating and mitigating language model biases against minoritized language varieties like African American Language (AAL), the dialects

---

[1]Our code is available at https://github.com/NickDeas/PhonATe

of English primarily associated with the African American community in the United States (Grieser, 2022). AAL, in particular, has minimal representation in both pre-training corpora for developing LLMs and finetuning data for tasks such as toxicity detection and sentiment analysis (e.g., see Xia et al. 2020). In contrast, text resources overwhelmingly reflect White Mainstream English (WME) , the variety of English primarily used by White Americans and reinforced as the norm of English (Baker-Bell, 2020)[2]. Reliable evaluation is vital to measuring and mitigating potential linguistic biases that can pose allocational (e.g., accessibility of LLMs) and representational (e.g., disproportionately labeling AAL as toxic) harms to historically marginalized language communities [3].

To account for this representation disparity, recent works have proposed data augmentation techniques through morphosyntactic and lexical transformations (Tan et al., 2020; Ziems et al., 2022; 2023). Other work has also traced biased predictions on tasks like toxicity detection to lexical features (Zhou et al., 2021). While these approaches have contributed to understanding of language model biases, they do not fully capture one particular class of these lexical features, which we refer to as *type-written phonological features*.

Type-written phonological features are spelling variants found in written text that reflect phonetic variation, such as writing *goin* in place of *going*. Past work has found that such features are primarily associated with texting, social media, and other digital contexts (Hillewaert, 2015; Cunningham, 2014; Ali et al., 2022). Furthermore, Eisenstein (2013) finds significant correlations on Twitter between the use of type-written phonological features commonly associated with AAL and county-level demographic data.

To improve understanding of models' behavior with AAL text found online and inform bias mitigation approaches, we focus on type-written phonological features of AAL and how these features impact model predictions. To do so, we introduce Phonological Augmentations for Text (PHONATE), employing models to convert between graphemes (characters in normal text) and phonemes (phonetic symbols) as well as a set of 10 rule-based phoneme transformations derived from sociolinguistic literature(Green, 2009; Bailey & Thomas, 2021; Thomas, 2007; Pollock et al., 1998). We then use this approach to augment existing datasets for tasks with type-written phonological features of AAL and study these features' effect on generative language models' behavior in a controlled manner.

| Task | Input Text | Prediction |
|---|---|---|
| Sentiment | done **with** finals, aka done **with** school, HOT JAM tonight,... | Neutral |
| | done **wif** finals, aka done **wif** school, HOT JAM tonight,... | Negative |
| Toxicity | I'm **afraid** you don't know what you're **talking** about! | Non-Toxic |
| | I'm **fraid** you dont no what you're **talkin** about! | Toxic |

Table 1: Example toxicity and sentiment predictions of Flan-T5 on WME text and text with synthetic type-written phonological features. Synthetic type-written phonological features (**bolded**) of AAL using PHONATE lead the model to predict a non-toxic text as toxic and a neutral text as expressing negative sentiment.

We emphasize that we develop this data augmentation approach strictly to evaluate and analyze model behavior and we do not advocate for developing models by training on synthetically generated AAL text (ethical implications are discussed further in section 6). We summarize our contributions as follows:

1. We introduce a novel approach, PHONATE, to generate synthetic typewritten phonological features for use in evaluating language model performance on dialectal variation.

---

[2]We acknowledge the different terms used to refer to these varieties and follow those used by Baker-Bell (2020) and Alim & Smitherman (2012).

[3]We follow Blodgett et al. (2020) in identifying harms. Allocational harms refer to the unfair distribution of resources among social groups, while representational harms refer to the inaccurate representation or unfair treatment of social groups. The harms we identify follow those of Deas et al. (2023), particularly in applications with high social impact.

2. We show that finetuning on text with PHONATE transformations reduces measured AAL biases across most models and tasks, suggesting that type-written phonological features are a vital consideration for bias mitigation approaches.
3. We find that particular phonological features can drastically alter model predictions on toxicity and sentiment tasks.

## 2  PHONATE Approach

While some orthographic variants may be predictable given a specific phonological feature, such as the use of g-dropping in spelling *going* as *goin*, many other features have more complex effects on orthography, such as the reduced diphthong in spelling *my* as *mah*. Impacts on orthography may be complicated further by the lack of a one-to-one correspondence between phonemes and graphemes in English (Fry, 2004; Thorndike & Lorge, 1944; Hanna et al., 1966). For example, the voicing distinction between the interdental fricatives (*th*) in *thing* (/θɪŋ/) and *the* (/ðə/) have different corresponding stops (/d/ and /t/ respectively) that may be realized in spelling as *da* and *ting*, capturing this distinction despite originally represented with the same grapheme.

The PHONATE augmentation process involves transcribing each text into a phoneme sequence, applying sociolinguistics-informed phonological rules, and transcribing the augmented phoneme sequence back into a new grapheme sequence. To better control the frequency of each feature, we incorporate hyperparameters designating the probability that each feature is applied.

**Phoneme/Grapheme Models.** To transcribe sentences between graphemes and phonemes, we employ an existing multilingual grapheme-to-phoneme (G2P) model and finetune a custom phoneme-to-grapheme (P2G) model on AAL data. The multilingual G2P model released by Zhu et al. (2022) transcribes text into IPA phonetic symbols. The model is based on byT5 (Xue et al., 2022), which processes and generates individual bytes rather than relying on a tokenizer, making it more suitable for character-level tasks. Compared to a pronunciation dictionary, this approach can also predict phoneme sequences for previously unseen words such as those that may appear on social media but are unlikely to be included in existing dictionaries.

To ensure that the P2G model used in PHONATE is able to generate words with type-written phonological features of AAL, we finetune a custom model. The training data includes both existing English pronunciation data originally used in training the G2P model[4] as well as all words in the TwitterAAE corpus (Blodgett et al., 2018) with corresponding phoneme sequences predicted by the G2P model. Further details on finetuning the P2G model are included in Appendix C.

**Phonological Rules.** We use a set of 10 phoneme transformation rules[5] derived from phonological features studied in sociolinguistics literature (Green, 2009; Bailey & Thomas, 2021; Thomas, 2007; Pollock et al., 1998). Each phonological feature is encoded using phonemic regex patterns, deleting phonemes or replacing phonemes with the appropriate alternative based on descriptions in literature.

For example, g-dropping is the orthographic feature[6] corresponding to the pronunciation of the velar nasal (e.g., the *-ng* in *running*) as an alveolar nasal (e.g., the *-n* in *runnin*) while monophthongization describes the simplification of diphthongs, which are two adjacent vowel sounds (e.g., the *y* in *my*) to a single vowel sound (e.g., the *-a* in *ma*). Based on these features, the corresponding regex-based transformations identify "η"s at word boundaries and replace them with "n"s to simulate g-dropping as well as remove the second vowel from sequences of two vowels to simulate monophthongization. In addition to g-dropping and monophthongization, we create transformation rules for r-lessness, l-deletion,

---

[4] `https://github.com/open-dict-data/ipa-dict`

[5] Each feature may be represented by multiple transformations depending on context, such as *th*-substitutions which depend on the location in the word as well as the original voicing.

[6] While the phonetic feature is not referred to in this way, we use g-dropping to refer to the orthographic realization.

*th-* substitutions, consonant cluster reduction, morpheme-final devoicing, *str-*backing, and stress-dropping. These are among the most common phonological features of AAL as well as among the most common to be realized as type-written phonological features. Descriptions of each phonological feature studied and the accompanying orthographic transformations are included in Appendix A and Appendix B respectively.

**Filtering.** Because the transformations operate on phoneme sequences, the transformations may inadvertently alter the meaning of the sentence such as converting past tense verbs to present tense. For example, *passed* in the original text may undergo the transformation for Monophthongization and become *pass*. To avoid these transformations, we use the spaCy part-of-speech (POS) tagger to label all original and augmented texts, and revert any transformations that result in a distinct recognized term or a term with a different predicted part of speech tag.

## 3 Methods

### 3.1 Tasks

Models are evaluated on three tasks: toxicity detection, sentiment analysis, and masked span prediction. Biases against AAL have been identified for both toxicity (e.g., Sap et al. 2019) and sentiment (e.g., Resende et al. 2024) classification tasks, but no studies to the authors' knowledge have examined these tasks in the context of AAL in generative language models. Additionally, prior work evaluating biases in generative language models have been limited to primarily intrinsic evaluations (Deas et al., 2023; Groenwold et al., 2020), while our work also includes extrinsic evaluations of model performance. We evaluate models on a masked span prediction (MSP) task following these prior works on generative models as well.

### 3.2 Data

| Task | Dataset | Dialect | Size | Avg. Length |
|---|---|---|---|---|
| Toxicity | DWMW17 (Davidson et al., 2017) Jigsaw | WME | 183,675 | 62.48 |
| Sentiment | TweetEval Sentiment (Barbieri et al., 2020) | | 45,614 | 19.35 |
| MSP | BookCorpus (Zhu et al., 2015) | | 20,000 | 11.20 |
| Evaluation | TwitterAAE subset (Groenwold et al., 2020) | AAL | 2,019 | 20.51 |

Table 2: Summary statistics for datasets used throughout experiments. Average length is measured in tokens using spaCy.

To finetune models, we use existing datasets primarily composed of WME and filter out potential AAL texts to form baselines with no additional exposure to AAL and to remove potentially biased annotations of AAL in the original datasets (Sap et al., 2022)[7]. We then augment these datasets using PHONATE to introduce synthetic type-written phonological features and compare against other data augmentations. To evaluate downstream performance, we use naturally occurring AAL to measure impacts on model predictions. All datasets used are summarized in Table 2.

**Toxicity Detection.** For toxicity detection experiments, *DWMW17* (Davidson et al., 2017) and *Jigsaw* [8] are used for finetuning all model variants. The datasets include samples of Tweets and Wikipedia comments respectively labelled for toxicity. We reduce the combined dataset to two labels, conserving "non-toxic" labels and collapsing all other to "toxic" (toxic, severely toxic, obscene, threat, insult, identity hate, offensive, hate speech, or abusive).

---

[7]Following (Xia et al., 2020), we use the classifier introduced in Blodgett et al. (2016) to remove all texts for which the predicted probability of representing AAL is higher than 0.8.

[8]https://www.kaggle.com/competitions/jigsaw-toxic-comment-classification-challenge/data

**Sentiment Analysis.** For the sentiment analysis experiments, we use the sentiment analysis subset of the *TweetEval* corpus (Barbieri et al., 2020) to finetune models. The corpus includes Tweets labeled for a variety of tasks including emotion detection, stance detection, and sentiment analysis. The sentiment analysis subset includes labels for each Tweet as either negative, neutral, or positive sentiment.

**Masked Span Prediction.** Finally, masked span prediction experiments use the *BookCorpus* dataset (Zhu et al., 2015) for finetuning. The BookCorpus includes short sentences from thousands of books by authors that were unpublished at the time.

**Evaluation.** To evaluate finetuned models on genuine AAL text on social media, we use a corpus of Tweets reflecting AAL: the *TwitterAAE* corpus Blodgett et al. (2018). Prior work has found that an extremely small percentage of randomly sampled Tweets are toxic (Founta et al., 2018). Based on this and following Xia et al. (2020), we also assume that all texts in TwitterAAE are non-toxic. In particular, we use the subset of TwitterAAE texts and the associated WME counterparts introduced in (Groenwold et al., 2020).

## 3.3 Metrics

**Augmentation Quality.** To evaluate the quality of the rule-based phonological transformations, we conduct a human evaluation of augmented texts. We create the evaluation data by first sampling 60 WME texts from the data introduced in Groenwold et al. (2020) and then apply PHONATE to the WME counterparts to generate pairs of WME and corresponding texts with synthetic type-written phonological features of AAL. As baselines, we also collect judgments for the original AAL counterparts in the dataset to compare the quality of PHONATE transformations to naturally occurring AAL texts. We ask annotators to rate each text on two dimensions using 5-point Likert scales. *Naturalness* asks how likely the text would be written by a human on social media; and following Deas et al. (2023), *Meaning Preservation* asks how well the augmented text conserves the meaning of the original text. Each annotator is asked to rate 50 texts such that 40 texts are shared between them. Ratings for these shared texts are averaged. Both annotators are self-identified AAL speakers studying linguistics.

**Toxicity.** To evaluate the impact of different data variants on toxicity bias measures, we examine the False Positive Rate (FPR) on the AAL test set. Using the assumption that a small fraction of the TwitterAAE (Blodgett et al., 2018) dataset is toxic, we consider any text predicted to be toxic as a false positive (Founta et al., 2018; Xia et al., 2020). In the experiments, FPR is calculated as $FPR = \frac{\# \, Toxic \, Predictions}{\# \, Texts}$, and we consider lower FPR percentages to indicate less biased model toxicity predictions.

**Sentiment Detection.** To evaluate how each data augmentation technique affects perceived sentiment biases, we again examine how often models trained on each dataset variant label a given text with negative sentiment. Unlike the toxicity task, sentiment expressed on social media is often mixed preventing the assumption that most texts express positive sentiment. Instead, we extract the pairs of AAL and WME text for which the model labels the WME text as positive sentiment and calculate the percentage of AAL texts in the subset that are labeled as negative sentiment. Within the set of positively labeled WME texts, the rate of negative predictions is calculated as $\frac{\# \, Negative \, AAL \, Predictions}{\# \, Positive \, WME \, Predictions}$. We consider lower percentages to illustrate less biased predictions as it suggests that models are less likely to label a positive sentiment text as negative due to the presence of AAL features.

**Masked Span Prediction.** Finally, we evaluate models in a generation setting where models are tasked with predicting a word or phrase masked from the input as in Deas et al. (2023). Performance on this task is measured with perplexity of the generated span to reflect how likely the original span is from the model's perspective, and top-k entropy to reflect the model's overall confidence in the predicted spans. Lower perplexities are considered to reflect less biased model behavior as lower scores suggest that the model gives higher probability to AAL texts.

## 3.4 Models

We evaluate the effect of type-written phonological features on four generative language models, expanding on prior work identifying AAL biases in generative language models (Deas et al., 2023; Groenwold et al., 2020). *T5 (large)* is an encoder-decoder model with 770 million parameters and is pre-trained on the C4 dataset as well as several generation tasks such as summarization (Raffel et al., 2020). With the same architecture, *Flan-T5 (large)* has 780 million parameters and is instruction-tuned on a larger set of tasks as well as chain-of-thought data (Chung et al., 2022). To represent larger, open-source models, we evaluate *Mistral-7B*, a 7.3 Billion parameter decoder-only model (Jiang et al., 2023). Because Mistral is a decoder-only model and is not pretrained on the same masked span prediction task as T5 and Flan-T5, we restrict its evaluation to the toxicity and sentiment tasks. Finally, we evaluate the robustness of ChatGPT, a commercially available, chat-based model trained on human preferences. Model checkpoints and prompts are included in Appendix D and details on finetuning and generation hyperparameters are included in Appendix E.

## 3.5 Experimental Setup

First, we evaluate the effect of finetuning on synthetic data. Measuring the impact of synthetic finetuning on real AAL data performance can suggest whether better representation of type-written phonological features or knowledge of these features can aid in debiasing language models on these tasks. To establish a baseline, models are finetuned on the original WME data with no augmentations (Base) to reflect models with no additional exposure to AAL. To represent the impact of morphosyntactic features alone, a set of models are finetuned on the data augmented using VALUE (Ziems et al., 2022). In this case, all features are applied to the text except the lexical features in order to isolate the impact of PHONATE on model predictions. Finally, a set of models are finetuned on data with morphosyntactic and random phonological augmentations to isolate the effect of systematic augmentations inspired by phonological features of AAL against random perturbations[9]. Finetuning hyperparameters for T5, Flan-T5, and Mistral are included in Appendix F.

We compare all baseline data variants to data augmented with both VALUE and PHONATE to reflect the added benefit of phonological features in finetuning data. Across all tasks, phonological transformations are performed where applicable with probability 20%. We stress that this does not accurately reflect the frequency or use of type-written phonological features online as our intent is not necessarily to produce realistic AAL text, but to study and evaluate models' behavior with these features. Examples of paired original texts and PHONATE-augmented texts are included in Table 1.

In a second set of experiments, we leverage PHONATE to investigate and better isolate the impact of each feature on model predictions. 5,000 texts with non-toxic labels for toxicity task and 5,000 texts with positive labels for the sentiment task are randomly sampled from the original dataset to analyze T5, Flan-T5, and Mistral. We examine 3 variants of the sample for each transformation: the original, VALUE-augmented, and PHONATE-augmented variants. Unlike in the finetuning experiments, each phonological feature is applied individually to the sample in order to isolate the effect of each feature[10]. We examine WME-finetuned models predictions on each variant measuring incorrect predictions. Due to the large number of texts for each feature, ChatGPT is evaluated on a smaller sample of 500 candidate texts.

---

[9]Vowels are only replaced with other vowels, and consonants only with other consonants. For a fair comparison, we restrict random augmentations such that the same counts of insertions, substitutions, and deletions align with the transformations applied using PHONATE for each text.

[10]Samples are subset to only the texts where it is possible to synthetically introduce each feature.

## 4 Results

### 4.1 PHONATE Quality

Table 4 shows the average ratings for the original AAL texts in (Groenwold et al., 2020) and WME texts augmented with PHONATE. Most PHONATE-augmented texts (61.9%) are rated 3 or higher for naturalness, suggesting that a majority of transformations reasonably reflect natural type-written phonological features. The Naturalness score for PHONATE falls below that of the human-written texts, but this is expected for synthetic augmentations, particularly as the judged PHONATE texts include type-written phonological features alone with no other features. Additionally, PHONATE receives a high rating for Meaning Preservation, suggesting that observed changes to model performance in the presence of PHONATE features are likely due to the orthographic changes rather than incidental meaning changes.

| Original WME Text | PHONATE-Augmented Text |
|---|---|
| I see now. Thanks for **clearing that** up. Talk | I see **na**. Thanks for **clearin dat** up. Talk |
| I **understand**. I'm the only one **left**. I'm going to check out Centiare too. Looks like it will be **way** better **than** WP. | I **understan**. I'm **deh** only one **lef**. I'ma check out Centiare too. Looks like it will be **wa** better **den** WP. |

Table 3: Example texts with PHONATE transformations (**bolded**) on all applicable terms.

| Texts | Natural | | Meaning | |
|---|---|---|---|---|
| | Avg. Score | % ≥ 3 | Avg. Score | % ≥ 3 |
| Human | 4.24 | 95.2% | 4.52 | 100% |
| PHONATE | 3.01 | 61.9% | 4.69 | 100% |

Table 4: Average ratings and percentage of ratings ≥ 3 for PHONATE transformations and original human-written AAL counterparts from the (Groenwold et al., 2020) dataset.

### 4.2 Synthetic Phonological Features in Finetuning

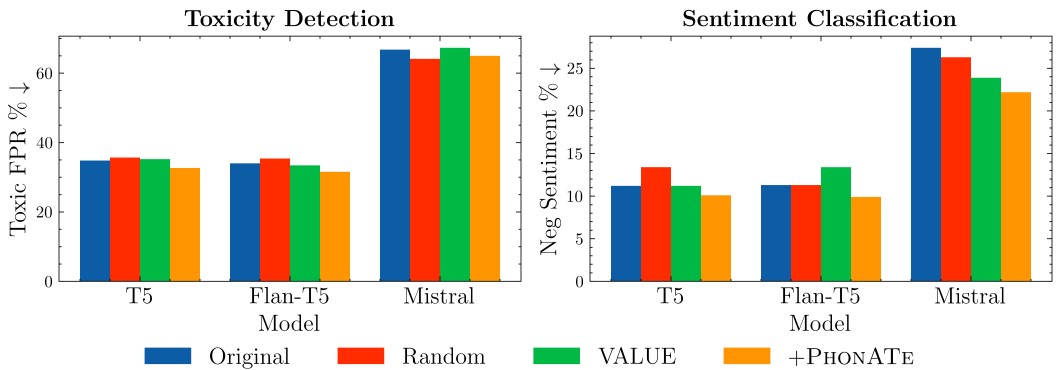

Figure 1: Toxicity False Positive Rates (left) and rate of negative sentiment predictions (right) on the TwitterAAE (Blodgett et al., 2018) corpus for different finetuning variants. Lower FPR and Negative Sentiment %s indicate less biased model predictions.

**Toxicity.** Figure 1 presents the FPRs for models finetuned on each dataset variant. First, T5 and Flan-T5 finetuned on random augmentations (*Random*) increase the FPR over the WME-finetuned baselines (*Original*), suggesting that random augmentations do not consistently improve model understanding or robustness to phonological features. Morphosyntactic augmentations (*VALUE*) also slightly raise the FPR for T5 and Mistral, but lower the FPR for Flan-T5, suggesting that morphosyntactic features alone do not consistently aid in toxicity detection tasks. Both morphosyntactic and phonological features (+PHONATE),

however, lower the FPR for all models, and lower the FPR the most for T5 and Flan-T5. This supports our hypothesis that phonological features are significant in toxicity detection models, as the use of systematic phonological features rather than random or morphosyntactic augmentations improves the FPR most.

**Sentiment Detection.** Figure 1 also includes the rate of negative sentiment labels on AAL texts where the WME counterparts are labeled with positive sentiment across finetuning dataset variants. As with the toxicity task, random augmentation (*Random*) raises the rate of negative sentiment labels for T5 and Flan-T5 compared to the WME-finetuned baseline (*Original*), while morphosyntactic augmentations (*VALUE*) alone slightly raise the rate for T5 and slightly lower the rate for Flan-T5 and Mistral. Also similar, finetuning with the combination of morphosyntactic and phonological augmentations (+PHONATE) results in the lowest rate of negative sentiment predictions across models, suggesting again that representation of phonological features is significant for sentiment analysis models predicting on AAL text as well.

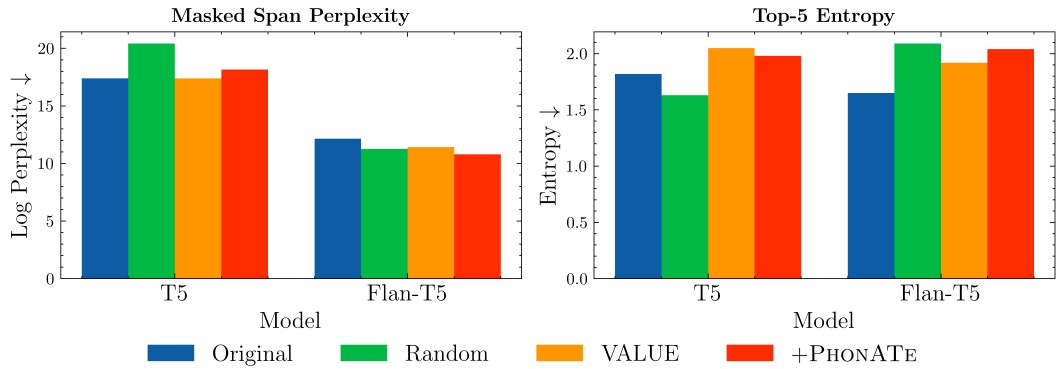

Figure 2: Log Perplexity and Top-5 Entropy scores on the TwitterAAE (Blodgett et al., 2018) corpus for T5 and Flan-T5 finetuning variants. Lower log perplexities indicate the model designates higher probability to masked AAL spans, while lower entropies indicate the model places higher probability on its top predictions.

**Masked Span Prediction.** Unlike the toxicity and sentiment tasks, the results for the masked span prediction task present less consistent trends, shown in Figure 2 (full results included in Appendix G). In this case, for T5, the WME dataset (*Original*) and morphosyntax-augmented dataset (*VALUE*) result in the lowest perplexity scores. At the same time, the same variants of Flan-T5 yield the highest perplexity scores and the lowest entropy scores, suggesting that they confidently predict incorrect spans on the AAL data. Despite not having the lowest perplexity for T5, the combination of morphosyntactic and phonological augmentations (+PHONATE) does yield the lowest perplexity for Flan-T5, consistent with the toxicity and sentiment tasks. Additionally, the addition of phonological transformations still exhibits a lower perplexity than random augmentations (*Random*) in both cases.

### 4.3   Model Robustness to Phonological Features

**Toxicity.** Figure 3 presents the FPRs when phonological features are applied in isolation alongside the average FPRs on the original and VALUE-augmented data (FPRs on original and VALUE-augmented data within each subset are shown in Appendix H). Overall, while certain features lead to minor changes in performance, others result in notably higher FPRs. In particular, consonant cluster reduction and final devoicing cause large increases in FPR for T5, Flan-T5, and Mistral, as well as monophthongization for T5 and Flan-T5. Interestingly, the most and least impactful features are highly similar between T5 and Flan-T5, possibly because instruction finetuning includes little or no AAL to impact performance. The most impactful features for ChatGPT are largely different from the other models, with predictions primarily impacted by features such as th-substitutions, r-lessness, and l-lessness. This may signal differences in the distribution of AAL features among models' pre-training corpora.

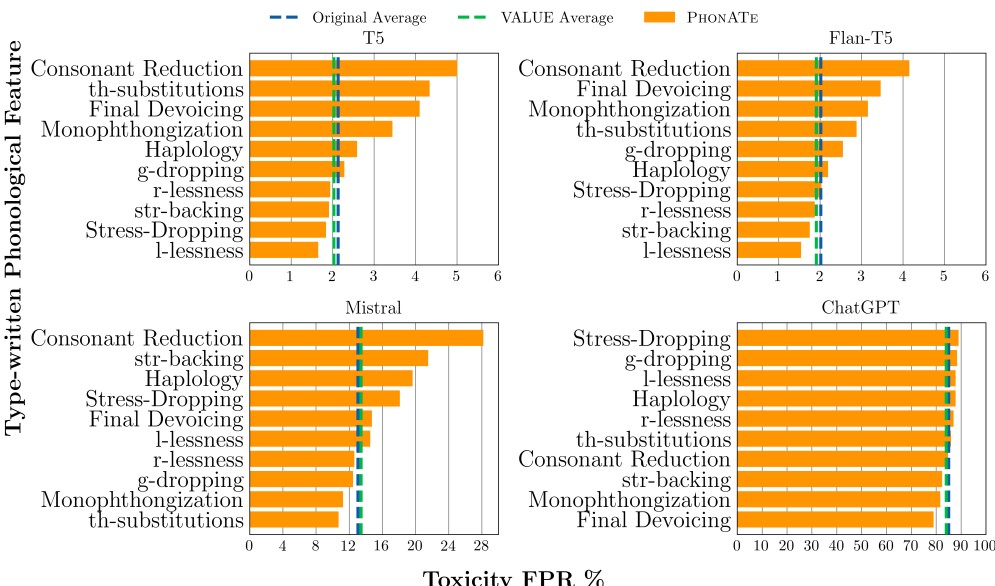

Figure 3: Rate of toxic predictions of WME-finetuned models on PHONATE-augmented non-toxic texts by type-written phonological feature. Average FPRs on original and VALUE-augmented texts are indicated by dashed lines. Type-written phonological features (y-axes) are ordered by largest impact, and higher FPRs indicate greater impact on model predictions.

**Sentiment.** A similar analysis for the sentiment task is shown in Figure 4, presenting the rate of Neutral or Negative sentiment labels for each phonological transformation alongside original and VALUE-augmented data variants. In contrast to the toxicity task, all phonological transformations cause notable increases in the rate of negative or neutral labels for T5 and Flan-T5. Again, the order of the most impactful features is largely similar between the two models, though the order largely differs from that of Mistral, ChatGPT, and the results in the toxicity analysis.

Between both the toxicity and sentiment robustness results, consonant cluster reduction, final devoicing, and monophthongization appear to be responsible for some of the largest increases in the rate of toxic or neutral/negative sentiment predictions for T5, Flan-T5, and Mistral. ChatGPT, however, appears to be more consistently impacted by stress dropping and th-substitutions in both tasks.

## 5 Related Works

**AAL Bias in NLP Tasks.** Models and datasets across a variety of tasks have been shown to be biased or less performant with AAL texts. One of the most studied settings for AAL biases is toxicity detection (Sap et al., 2019; 2022; Chuang et al., 2021; Xia et al., 2020). Particularly related to the impact of phonological features and lexical variants, Zhou et al. (2021) examines how lexical items and dialect markers of AAL may trigger toxicity systems to falsely label a text as toxic. In addition to toxicity detection, Resende et al. (2024) also identifies how sentiment analysis systems are disproportionately likely to label African American English expressions with negative sentiment. Beyond these tasks, performance disparities have also been uncovered in Automatic Speech Recognition systems (Martin & Tang, 2020; Koenecke et al., 2020; Mengesha et al., 2021), language identification systems (Blodgett et al., 2016; Blodgett & O'Connor, 2017), and models for dependency-parsing and POS-tagging (Blodgett et al., 2018; Jørgensen et al., 2016). Biases in generative language models, however, have received little attention and been constrained to intrinsic evaluations (Deas et al., 2023; Groenwold et al., 2020). In contrast, we include extrinsic evaluations

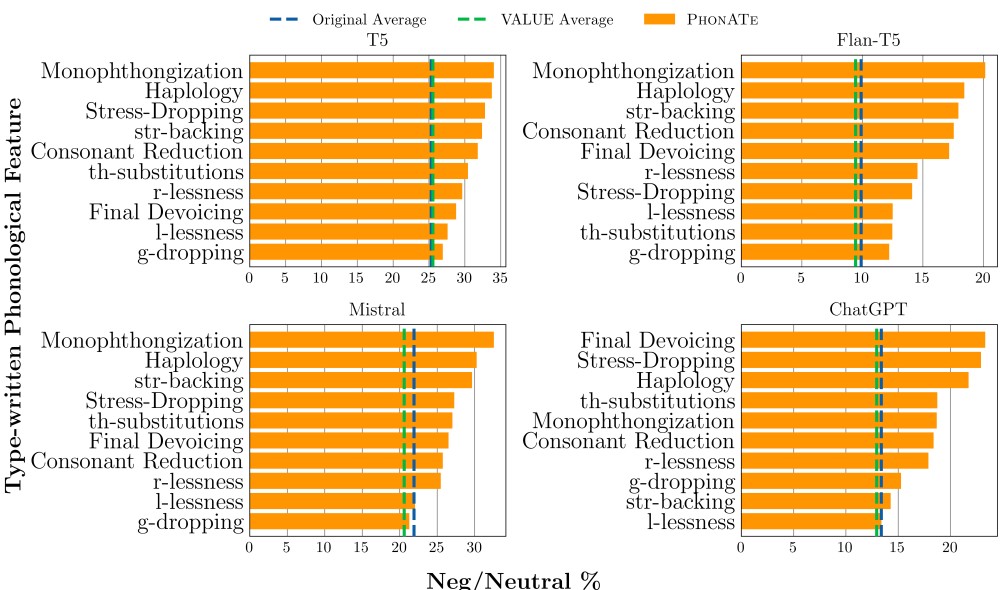

Figure 4: Rate of negative or neutral predictions of WME-finetuned models on PHONATE-augmented, positive sentiment texts by type-written phonological feature. Average negative/neutral rates on original and VALUE-augmented texts are indicated by dashed lines. Type-written phonological features (y-axes) are ordered by largest impact, and higher %s indicate a greater impact on model predictions.

of models on toxicity detection and sentiment analysis tasks while also giving focus to type-written phonological features.

**Linguistic Data Augmentation.** Several prior works have used data augmentation for both evaluating and mitigating societal biases in language models. Maudslay et al. (2019) augments data with counterfactuals and name substitutions to mitigate gender biases, while Qian et al. (2022) uses neural perturbations to augment data along multiple demographic axes. Similar approaches have been applied to linguistic biases as well, using augmentations to approximate linguistic features of a language variety or dialect. Morpheus (Tan et al., 2020) augments word inflections adversarial during training to make models more robust to L2 English and AAL speakers. Alternatively, to better reflect the targeted language variety, Ziems et al. (2022) and Ziems et al. (2023) use surface-level transformations to synthetically introduce morphosyntactic and lexical features of a variety of dialects. In contrast to these approaches, we study linguistic biases against AAL focusing on synthetically introducing type-written phonological features without relying on limited dictionaries.

## 6 Conclusion

We introduce a novel approach, PHONATE, to augment WME texts with synthetic type-written phonological features of AAL. We use PHONATE to conduct two sets of experiments evaluating the significance of type-written phonological features on generative language models in three tasks. First, we finetune models for toxicity, sentiment, and masked span prediction using datasets with different sets of augmentations. We find that finetuning with both morphosyntactic and phonological features typically results in the best or least biased performance on toxicity, sentiment, and masked span prediction tasks, outperforming finetuning with morphosyntactic features alone and random phonological augmentations. Additionally, we evaluate how often morphosyntactic and individual phonological features alter model predictions on the toxicity and sentiment tasks, finding that particular phonological features have large impacts on model predictions.

## Limitations

We recognize several limitations accompanying the results of our experiments. First, the augmentations applied by PHONATE are not exhaustive and do not perfectly capture the natural use of type-written phonological features. The aim in using the PHONATE approach is to improve the recall of type-written phonological features by avoiding reliance on a finite list of features, and we combat potential inaccuracies by filtering transformations with POS-tagging. Additionally, both PHONATE and dictionary approaches to incorporating type-written phonological features are limited in that they do not consider larger context in applying transformations (e.g., some features depend on the preceding or following word). As there are few studies of the use of type-written phonological features, we leave improved augmentations and study of the differences between spoken and written phonological features to future work.

Additionally, we limit our evaluation to generative models on three tasks–toxicity, sentiment, and masked span prediction–which may not reflect the same patterns as with other tasks. At the same time, in the masked span prediction experiments, we are unable to reliably mask only AAL features when evaluating models, leading the scores to also reflect performance on spans without features. We examine these tasks based on prior studies that have uncovered biases in both encoder-only and generative language models as well as to include both an extrinsic and intrinsic evaluation of the included models. We leave further analysis, evaluation of classification model architectures, and evaluation on other tasks to future work.

## Ethics Statement

We recognize that in developing an approach to simulate features of African American Language, it could be used to finetune a model to generate AAL and mimic AAL speakers online. To avoid this in our own experiments, we refrain from finetuning any model to *generate* AAL using the synthetic augmentations: in the toxicity and sentiment experiments, only the inputs are modified, and in the masked span prediction task, we avoid masking augmented tokens. Additionally, we emphasize that finetuning on data augmented to introduce synthetic features of AAL is not intended as an approach to mitigate, and our primary aim is to show that phonological features in online texts have a notable impact on model performance. We are committed to openness and transparency in our research on the impact of AAL features on LLM performance and our ultimate aim is the development of core mitigation strategies that can address problems in current practice. Given the historical linguistic subordination of AAL that derives from prejudices against it speakers, we feel it is imperative to center AAL speakers in decision-making processes, and thus, we recruit AAE-speakers to validate our approach. The research must come out of Black communities in order to develop best practices around the use of AAL data and the development of applications with social impact that must be made available to all communities.

## Acknowledgements

This work was supported in part by grant IIS-2106666 from the National Science Foundation, National Science Foundation Graduate Research Fellowship DGE-2036197, the Columbia University Provost Diversity Fellowship, and the Columbia School of Engineering and Applied Sciences Presidential Fellowship. Any opinion, findings, and conclusions or recommendations expressed in this material are those of the authors and do not necessarily reflect the views of the National Science Foundation.

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

## A  Phonological Features of AAL

Beyond morphosyntactic and lexical features, AAL also exhibits phonological and phonetic features that are realized as variation in the pronunciation of consonants and vowels in particular contexts (Thomas, 2007). Frequent features of AAL phonology considered in this work are detailed below, compiled from linguistic studies of AAL phonology (Green, 2009; Bailey & Thomas, 2021; Thomas, 2007; Pollock et al., 1998).

*r-lessness*, *non-rhoticity*, or *r-deletion* describes cases where the syllable and often word-final /r/ may not be pronounced or vocalized as /w/. In non-rhotic dialects, a word such as *teacher* (/ˈtitʃə/) may be pronounced as *teacha* (/ˈtitʃər/), or *store* (/stɔr/) as *sto* (/stɔw/).

*l-deletion* similarly involves deletion of /l/ sounds, in word-final positions or preceding labial consonants. The /l/ may be deleted entirely, or vocalized as /o/, /w/, or /ə/. For example, *cool* (/kul/) may be pronounced as *coo* (/ku/).

*g-dropping* occurs when the word-final *-ing* (-/ɪŋ/) may be pronounced as *-in* (/ɪn/) in non-monosyllabic words. For example, *running* (/rʌnɪŋ/) may be pronounced as *runnin* (/rʌnɪn/)

*th-fronting* alters the pronunciation of /θ/ (voiceless in ba*th*) and /ð/ (voiced in *th*at), with differing realizations depending on the position in the word and voicing. /θ/ may be pronounced as /t/ in word-initial positions or /f/ in word-final positions while /ð/ may be pronounced as /d/ in word-initial positions or /v/ in word-final positions. For example, *bath* (/baθ/) may be pronounced as *baf* (/baf/) while *that* (/ðat/) may be pronounced *dat* (/dat/).

*Consonant cluster reduction* simplifies clusters of two or more adjacent consonants in word-final positions. In most cases, the final consonant sound is deleted, such as how *first* (/fɜrst/) may be pronounced *firs* (/fɜrs/).

*Final devoicing* replaces morpheme-final voiced consonants with their voiceless counterparts. For example, *kids* (/kɪds/) may be pronounced as *kits* (/kɪts/).

A feature unique to AAL among North American dialects is *str-backing*, where /str/ in word-initial positions may be pronounced /skr/. For example, *street* (/strit/) may be pronounced *skreet* (/skrit/).

Beyond consonantal features, vowels may be altered through *monophthongization* or *diphthong simplification*, where diphthongs, two successive vowel sounds, are simplified to a monophthong, a single vowel sound. For example, *my* (/mai/) may be pronounced as *mah* (/ma/).

Finally, *stress-dropping* describes how word-initial, unstressed syllables may be deleted. A common example is realized when *about* (/əˈboʊt/) is pronounced *'bout* (/boʊt/).

## B  Rule-Based Phonological Transformations

Table 5 shows each feature with a description of the associated regex transformation. We employ this approach rather than an end-to-end approach to ensure adherence to the features as identified by sociolinguistics studies. This approach also better ensures that the meaning of the original text is conserved, given that prior work finds that style transfer approaches often fail to conserve meaning (Ziems et al., 2022).

## C  Phoneme-to-Grapheme Model finetuning

While we use an existing Grapheme-to-Phoneme model for transcribing text into phoneme sequences, PHONATE requires a Phoneme-to-Grapheme model capable of producing terms reflecting type-written phonological features of AAL. To do this, we create a dataset consisting of 1) the CMU English Pronouncing Dictionary[11] with added syllable markers[12], and 2) predicted pronunciations of AAL text on social media from Blodgett et al. (2018).

We use the multilingual Grapheme-to-Phoneme model Zhu et al. (2022) to generate predicted pronunciations of all terms in the TwitterAAE corpus. We filter out terms containing 3 or more sequential repeated characters (e.g., *yesssss*) or non-Latin characters before generating the paired AAL terms and predicted pronunciations.

---

[11]http://www.speech.cs.cmu.edu/cgi-bin/cmudict
[12]https://github.com/open-dict-data/ipa-dict

| Feature | Description | Phonemes/Patterns | Transformation |
|---|---|---|---|
| Monophthong-ization | diphthongs (sequences of 2 vowel sounds) are simplified to single monophthongs | diphthongs (i.e. /aʊ/ or /aɪ/) | First vowel phoneme in pair |
| r-lessness/r-deletion | /r/ phonemes at the ends of words or after consonants at beginnings of words can be dropped | /(rhotic)/ before consonant or after word-initial consonants | // |
| str-backing | [str] sounds pronounced [skr] | /st(rhotic)/ | /sk(rhotic)/ |
| th-substitutions | [th] sound pronounced as /t/, /f/, /d/, /fv depending on voicing and position | /θ/ or /ð/ | /d/, /t/ or /v/, /f/ |
| l-deletion | [l] sound deleted before labial consonants | /ll/ before labial consonant | labial consonant |
| Final devoicing | Substitute voiced final consonant with voiceless counterpart | /(voiced consonant)/ | /(devoiced consonant)/ |
| Haplology | Delete repeated vowel or syllable | repeated vowel or syllable | single instance of vowel or syllable |
| Consonant Cluster Reduction | Remove last consonant in word-final cluster of consonants | /(consonant){1,}(consonant)/ | /(any)(consonant){1,}/ |
| g-dropping | [ing] word endings replaced with [in] | /ɪŋ/ | /ɪn/ |
| Stress-dropping | Remove unstressed syllable at beginning of word | /(any){,3}ˈ(any)/ | /ˈ(any)/ |

Table 5: List of phonological rules derived from sociolinguistic literature on African American Language phonological features

With this data, we finetune a *byT5-small* model Xue et al. (2022) to produce the original term given the phoneme sequence reflecting pronunciation. The model is finetuned for a single epoch with learning rate 3e-4, linear learning rate schedule, and weight decay 0.01.

## D  Model Checkpoints and Prompts

For T5, Flan-T5, and Mistral, we use the *google-t5/t5-large*, *google/flan-t5*, and *mistralai/Mistral-7B-v0.1* checkpoints respectively made available through Huggingface. For querying Chat-GPT, we use the *gpt-3.5-turbo-1106* checkpoint for reproducibility and a temperature of 1 in generations.

Prompts used for each task and model are shown in Table 6. All prompts are drawn from (Si et al., 2023) and (Zhang et al., 2024). Mistral prompts are appended with an end token ("⟨/s⟩").

## E  Generation Hyperparameters

For toxicity and sentiment tasks, models are restricted to generating 1 and 3 tokens as expected outputs are ("Yes", "No") and ("Negative", "Neutral", "Positive") respectively. For the masked span prediction task, models can generate up to 64 tokens. All generation is performed with greedy decoding.

For reproducibility, we query ChatGPT with temperature 0 and seed 1. Only user messages are provided through the API with no system message.

| Task | Models | Prompt |
|---|---|---|
| Toxicity | T5, Flan-T5, Mistral | "Categorize the comment on the basis of toxicity. If the comment is toxic (harmful, malicious, derogatory, threat, insult, identity attack, etc.) output Yes, otherwise output No. [INPUT TEXT]" |
| | ChatGPT | "Categorize the comment on the basis of toxicity. If the comment is toxic (harmful, malicious, derogatory, threat, insult, identity attack, etc.) output Yes, otherwise output No. Sentence: [INPUT TEXT] Label:" |
| Sentiment | T5, Flan-T5, Mistral | "[INPUT TEXT] Sentiment: " |
| | ChatGPT | "Please perform Sentiment Classification task. Given the sentence, assign a sentiment label from ['Negative', 'Neutral', 'Positive']. Return label only without any other text. Sentence: [INPUT TEXT] Label:" |

Table 6: Prompts used for each model in Toxicity and Sentiment task finetuning and evaluation.

## F  Finetuning Hyperparameters

Across tasks, models are finetuned with identical hyperparameters, which are detailed below:

**T5 and Flan-T5** are finetuned with learning rate 1e-3, linear learning rate schedule, weight decay 0.01, and a batch size of 32.

**Mistral** is finetuned using LoRa (Hu et al., 2021) with flash attention. For LoRa, we use LoRa rank 64, LoRa alpha 16, LoRa dropout 0.05. Additionally, we use a maximum learning rate of 1e-4, minimum learning rate 2e-5, learning rate gamma 0.8, a cosine annealing learning rate schedule, weight decay .01, and batch size of 32, with 10 warmup steps and 50 cycle steps.

## G  Full Finetuning Results

Raw scores for the Toxicity and Sentiment tasks are shown in Table 7a, while raw perplexity and entropy values for the masked span prediction task are shown in Table 7b.

| Training Variant | Toxic FPR % | | | Neg Sentiment % | | |
|---|---|---|---|---|---|---|
| | T5 | Flan | Mistral | T5 | Flan | Mistral |
| Base | 34.8 | 34.0 | 66.8 | 11.2 | 11.3 | 27.4 |
| Random | 35.7 | 35.4 | **64.1** | 13.4 | 11.3 | 26.3 |
| VALUE | 35.2 | 33.4 | 67.3 | 11.2 | 13.4 | 23.9 |
| +PHONATE | **32.7** | **31.6** | 65.0 | **10.1** | **9.9** | **22.2** |

(a) False Positive Rates (FPR) and rate of negative sentiment predictions on the TwitterAAE (Blodgett et al., 2018) corpus for different finetuning variants.

| Training Variant | Model | | | |
|---|---|---|---|---|
| | T5 | | Flan | |
| | Perp | Ent | Perp | Ent |
| Base | **3.6e7** | 1.82 | 1.9e5 | 1.65 |
| Random | 7.4e8 | 1.63 | 7.8e4 | 2.09 |
| VALUE | **3.6e7** | 2.05 | 9.1e4 | 1.92 |
| +PHONATE | 7.7e7 | 1.98 | **4.9e4** | 2.04 |

(b) Raw perplexity and top-5 entropy scores on the TwitterAAE (Blodgett et al., 2018) corpus for T5 and Flan-T5 finetuning variants.

Table 7: Raw synthetic finetuning results for the Toxicity and Sentiment tasks (a) and Masked Span Prediction task (b).

## H    Full Robustness Results

Figure 5 and Figure 6 present the robustness results including the raw scores on the original and VALUE-augmented texts for each subset. As the percentages for each feature are calculated based on the subset of texts where a given feature may be applied, there is slight variance in the scores on the original and VALUE-augmented texts.

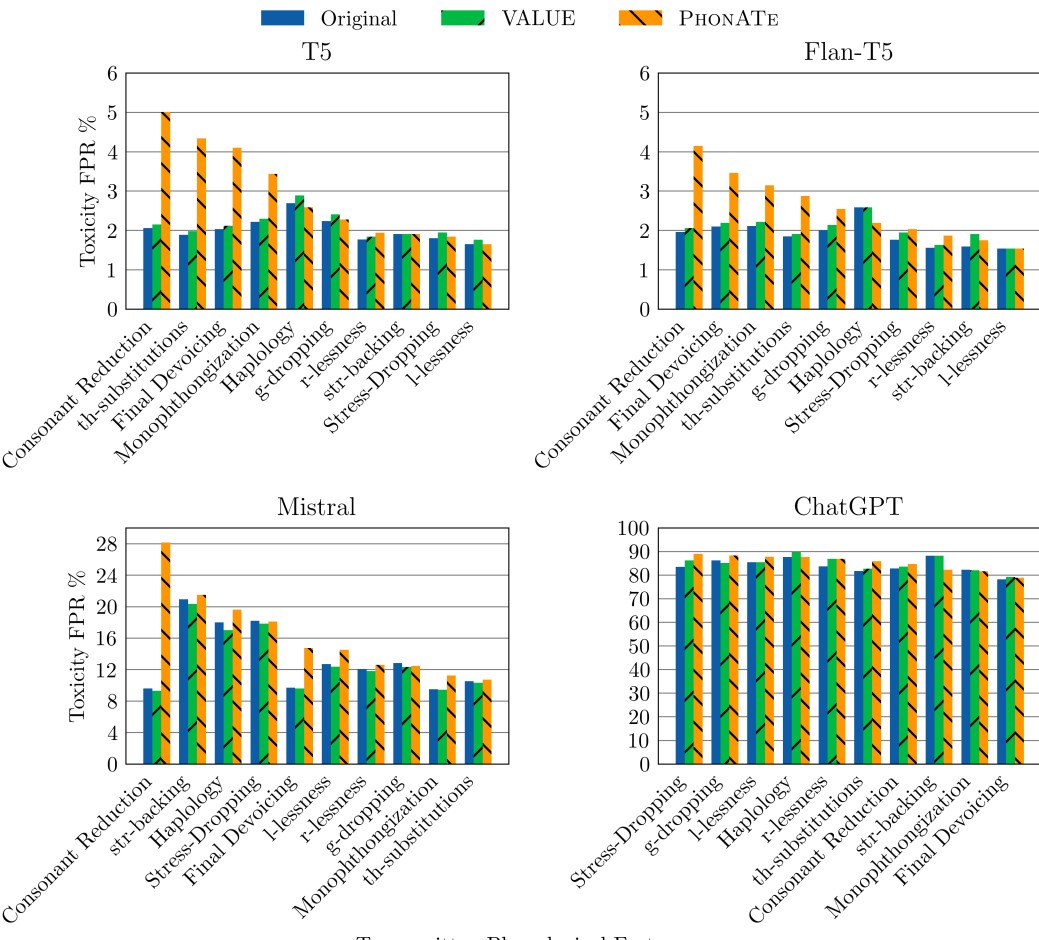

Figure 5: Rate of toxic predictions by type-written phonological feature of the WME-finetuned T5, Flan-T5, and Mistral models as well as ChatGPT on PHONATE-augmented non-toxic samples from the DWMW17 and Jigsaw datasets. FPR %s are shown as raw percentages for each data subset. Features (y-axis) are sorted by largest FPR on PHONATE-augmented data.

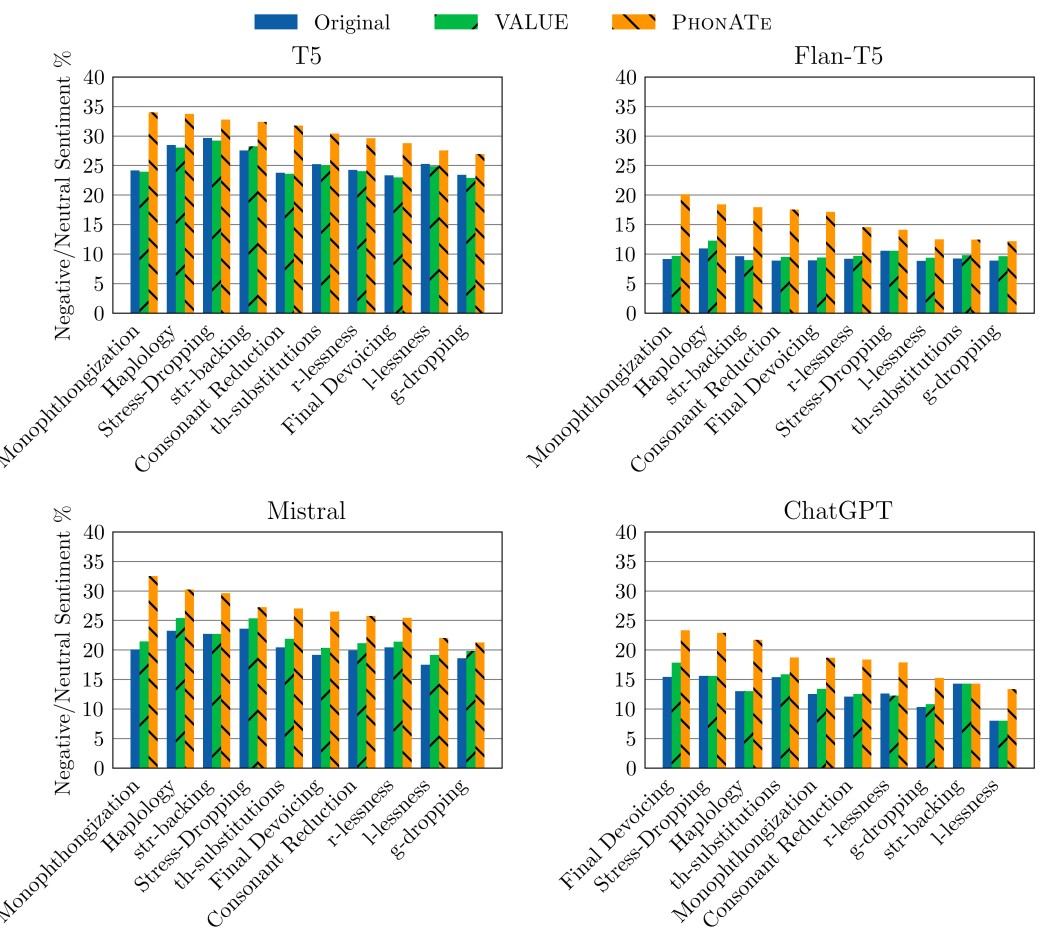

Figure 6: Rate of negative or neutral sentiment predictions by type-written phonological feature of the WME-finetuned T5, Flan-T5, and Mistral models as well as ChatGPT on PHONATE-augmented non-toxic samples from the TweetEval sentiment dataset. Negative/Neutral %s are shown as raw percentages for each data subset. Features (y-axis) are sorted by largest Negative/Neutral Sentiment % on PHONATE-augmented data.

