# OpenReview forum: "PhonATe: Impact of Type-Written Phonological Features of African American Language on Generative Language Modeling Tasks"
_colmweb.org/COLM/2024/Conference — COLM_

### Official Review · Reviewer_XUsW · 2024-05-02

**Rating:** 6
**Confidence:** 4
**Ethics Flag:** 1

**Summary:**

The paper presents an approach to enhance the performance of LLMs on texts with African American Language (AAL) features particularly focusing on type-written phonological features. They introduce an approach of generating synthetic typewritten phonological features that generates text with phonological transformations from a standard text. The paper investigates how injecting this synthetic AAL text into training datasets impacts the performance of LLMs on tasks such as toxicity detection, sentiment analysis, and masked span prediction.

**Reasons To Accept:**

The paper is very well written and the task is clearly explained. The idea of extending LLM data with synthetic ones is very interesting and I think this is an approach that could be extended to other related areas.The ablation study clearly shows what contributes to the improvements.

**Reasons To Reject:**

Perhaps the authors should have explained better why these phonological features make their way to text, no where in the paper transcription is mentioned, so I assume (from the cited papers) most of the time people write these phonologically transformed text on social media.
I understand that the authors have resource constraints but the two main LLMs used in the experiments are quite old and even Mistral is considered a small model. One cannot generalize these results and assume larger language models would behave similarly.
Apart from that, I really think this paper fits better in a conference like *CL as the linguistic content is quite rich but the LLM part is quite limited.

---

> ### Author Rebuttal · Authors · 2024-05-31
>
> Thank you for your review.
>
> __Reason to Reject 1:__ _Perhaps the authors should have explained better why these phonological features make their way to text, no where in the paper transcription is mentioned, so I assume (from the cited papers) most of the time people write these phonologically transformed text on social media._
>
> The reviewer is correct that most of the time these type-written phonological features are written by individuals and primarily occur on social media (paragraph 3 of Introduction). Our work focuses on the use of these features in written posts and does not use speech transcription. To our knowledge, there is little sociolinguistics research into why the spoken phonological features may affect orthography (paragraph 1 of Limitations), and we believe studying these reasons is outside of the scope of this work. We can, however, make more explicit that this research gap exists and emphasize that these features are primarily written by users on social media.
>
> __Reason to Reject 2:__ _I understand that the authors have resource constraints but the two main LLMs used in the experiments are quite old and even Mistral is considered a small model. One cannot generalize these results and assume larger language models would behave similarly. Apart from that, I really think this paper fits better in a conference like *CL as the linguistic content is quite rich but the LLM part is quite limited._
>
> Thank you for your suggestion. We are happy to evaluate a larger LLM in our robustness experiments. Additionally, we note that the scope of COLM is not just restricted to LLMs but language modeling more broadly, and we believe that the ethics and linguistics focus of our work falls within the Societal implications and LMs for everyone topics listed in the Call for Papers.

---

### Official Review · Reviewer_JF8K · 2024-05-10

**Rating:** 6
**Confidence:** 4
**Ethics Flag:** 1

**Summary:**

The authors study if introducing synthetic data mimicking African-American English dialects into model training can narrow the gap in performance on AAE evaluation sets. They first develop a framework to transform text into AAE based on typed phonological features of AAE derived from sociolinguistics literature. They then present results showing that finetuning on this synthetic language lowers false positive rates in toxicity detection with AAE and leads to fewer model predictions of negative sentiment on AAE text. They also conduct a masked span prediction experiment where they find mixed results in terms of perplexity after synthetic AAE finetuning, though it can lower perplexity in some cases.

**Questions To Authors:**

1. What was the error rate in the phonological transformations?

2. Why do you think finetuning works for lowering perplexity with Flan-T5 and not T5?

3. Figures 2/3 are hard to decipher

**Reasons To Accept:**

- I found this to be an interesting paper, which tackles a relatively known problem in NLP (lack of robustness to AAE) with an approach that is novel within this area (synthetically introducing phonological features). This is a important area of AI ethics, with recent work finding widespread LLM discrimination against AAE speakers, even leading to a higher likelihood of AAE speakers being sentenced to death by LLM judges [1].

- The authors do an excellent job in both the main body of the paper, limitations and ethics statement to address potential concerns around mimicking actual AAE speakers. I thought the ethics statement in particular was very thoughtful, calling for representation of affected communities in LLM application development.

- They analyze performance across multiple tasks and models. While I found there to be flaws in evaluation (see reasons to reject), I did think that their breakdown of effects on task performance from each of the phonological features was thorough, and potentially useful for informing future research on safe/non-discriminatory deployment of LLMs.

- I appreciated that they conduct a human evaluation to verify the quality of synthetic data with AAE speakers, which absolutely should be done in a study like this

[1] Hofmann, Valentin, Pratyusha Kalluri, Dan Jurafsky and Sharese King. “Dialect prejudice predicts AI decisions about people's character, employability, and criminality.” ArXiv abs/2403.00742 (2024).

**Reasons To Reject:**

- The writing is unclear in many places and should be revised before publication. For example, in the section on augmentation quality it states “both annotators are self-identified AAL speakers,” but it isn't stated explicitly that there are two annotators judging each example and how the final rating is derived. The description of the data subsets used on pg 6 ("5000 non-toxic texts and 5000 texts for the sentiment dataset") is also confusing.

- The sentiment detection eval seems very odd, given that there is no ground-truth for the sentiment. Presumably if an example is labeled positive in WME, it shouldn't become negative just from a transformation into AAE, however this experiment could be conducted in a cleaner way to make it clear the outcome is a result of bias.

-   The paper never really motivates the use of synthetic AAE instead of finetuning on a larger corpus of real AAE

---

> ### Author Rebuttal · Authors · 2024-05-31
>
> Thank you for your review.
>
> __Reason to Reject 1:__ _The writing is unclear in many places and should be revised before publication…_
>
> Thank you for pointing out places that could clarified. Each annotator is asked to rate 50 augmented and natural texts such that 40 of each overlap. For these 40, the ratings are averaged. Concerning data subsets, we sample 5,000 texts with non-toxic labels from the toxicity datasets as well as 5,000 texts with positive labels from the sentiment dataset. We will update the descriptions to clarify.
>
> __Reason to Reject 2:__ _The sentiment detection eval seems very odd, given that there is no ground-truth for the sentiment…_
>
> We directly measure the percentage of cases where the sentiment prediction does change and thus, we measure bias via this process. Prior work on other biases in sentiment detection similarly use differences in sentiment predictions (e.g., Goldfarb-Tarrant et al., 2023). We will more explicitly state why we consider this behavior biased and would be open to suggestions on improving the experiment.
>
> __Reason to Reject 3:__ _The paper never really motivates the use of synthetic AAE instead of finetuning on a larger corpus of real AAE_
>
> We note in the beginning of the Introduction that AAL has minimal representation in common datasets. While the TwitterAAE corpus is large, our aim was to better isolate the impacts of specific phonological features (e.g. the robustness experiments). We will more thoroughly motivate using synthetic data in our controlled experiments in the revision.
>
> __Question 1:__ _What was the error rate in the phonological transformations?_
>
> With multiple ways an author may spell a term reflecting phonological features, there are not necessarily correct or incorrect augmentations in all cases. Instead, we evaluate how natural the augmentations appear to AAL-speakers (percentages shown in Table 4).
>
> __Question 2:__ _Why do you think finetuning works for lowering perplexity with Flan-T5 and not T5?_
>
> As we cannot reliably mask only AAL features, model losses rise for some spans without features. For T5 in particular, a small set of samples have a large rise, but the additional pre-training of Flan-T5 may mitigate these effects. We will expand our Results and Limitations sections to discuss this behavior and the limitation of random masking.
>
> __Question 3:__ _Figures 2/3 are hard to decipher_
>
> Thank you for bringing this to our attention. We will make the figures clearer in the revision.

---

> > ### Comment · Reviewer_JF8K · 2024-06-05
> > **Rebuttal Response**
> >
> > Thank you the authors for responding to my concerns and questions. I look forward to the revised version and my review is unchanged.

---

### Official Review · Reviewer_7yF2 · 2024-05-10

**Rating:** 6
**Confidence:** 3
**Ethics Flag:** 1

**Summary:**

The study introduces Phonological Augmentations for Text (PhonATe), a method that converts between graphemes and phonemes, as well as rule-based phoneme transformations derived from sociolinguistic literature. By augmenting datasets with AAL phonological features, the researchers aim to understand how these features influence model predictions in tasks such as toxicity detection and sentiment analysis. The study also discusses the potential implications of incorporating AAL features into language models for bias mitigation and highlights the importance of developing more inclusive and accurate models for diverse linguistic contexts.

**Reasons To Accept:**

The paper introduces a novel approach, PhonATe, to synthetically introduce type-written phonological features of African American Language (AAL) into text, addressing a gap in prior research . This innovative method has the potential to enhance understanding of model behavior with AAL text and inform bias mitigation strategies in language models.

The paper demonstrates a commitment to openness, transparency, and ethical research practices in exploring the impact of AAL features on language model performance.

By focusing on type-written phonological features of AAL and their impact on model predictions, the study provides valuable insights that can inform the development of more inclusive and accurate language models.

**Reasons To Reject:**

While the paper states that fine-tuning on data augmented with synthetic AAL features is not intended as an approach to mitigate biases , there may be ambiguity in how the fine-tuning process was conducted and its implications on model behavior. Clearer explanations or guidelines on the fine-tuning methodology and its impact on model performance are necessary for transparency and reproducibility.

The paper emphasizes the importance of centering AAL speakers in decision-making processes . However, there is a lack of clarity on how the research engages with or involves the AAL community in the development and evaluation of the PhonATe approach. Involving relevant communities in the research process can provide valuable insights and ensure that the study is culturally sensitive and beneficial to those it aims to represent.

Previous research has highlighted how annotator beliefs and identities can bias toxic language detection tasks . Considering the sensitive nature of the tasks evaluated in the study (toxicity detection, sentiment analysis), there should be a thorough discussion on how potential biases in annotation or data collection were addressed to ensure the reliability and fairness of the results.

---

> ### Author Rebuttal · Authors · 2024-05-31
>
> Thank you for your review.
>
> __Reason to Reject 1:__ _While the paper states that fine-tuning on data augmented with synthetic AAL features is not intended as an approach to mitigate biases, there may be ambiguity in how the fine-tuning process was conducted and its implications on model behavior. Clearer explanations or guidelines on the fine-tuning methodology and its impact on model performance are necessary for transparency and reproducibility._
>
> All model checkpoints, prompts, generation hyperparameters, and fine-tuning hyperparameters are described in Appendices D-F (we direct readers to the Appendix in section 3.4). Thus, we include sufficient details for reproducing our experiments, and plan to release our code. If there are particular details the reviewer feels are missing, we will include them in the revision.
>
> __Reason to Reject 2:__ _The paper emphasizes the importance of centering AAL speakers in decision-making processes . However, there is a lack of clarity on how the research engages with or involves the AAL community in the development and evaluation of the PhonATe approach. Involving relevant communities in the research process can provide valuable insights and ensure that the study is culturally sensitive and beneficial to those it aims to represent._
>
> The development of PhonATe is drawn from sociolinguistic studies of AAL conducted with AAL-speakers. Some of the authors of this work are from the AAL community, including sociolinguistics researchers. Additionally, we note in the paper that the PhonATe evaluation is conducted by 2 AAL-speakers (“Augmentation Quality” on pg.4-5). We will further elaborate on how the AAL community was involved in this work in the Ethics statement.
>
> __Reason to Reject 3:__ _... Considering the sensitive nature of the tasks evaluated in the study (toxicity detection, sentiment analysis), there should be a thorough discussion on how potential biases in annotation or data collection were addressed to ensure the reliability and fairness of the results._
>
> We do not collect any data and instead rely on existing fine-tuning corpora. To mitigate the impacts of annotators beliefs and avoid relying on potentially biased AAL annotations, we use the demographic-alignment classifier introduced by Blodgett et al (2016) to filter out all texts  from the fine-tuning datasets predicted to reflect AAL given that they may have biased annotations (Paragraph 1 of Data section). We will elaborate further on this in the revision.

---

### Decision · Program_Chairs · 2024-07-10

**Decision:**

Accept

**Comment:**

This work aims to improve known issues of biases based on African American Language text in areas such as sentiment and toxicity (they also include masked span prediction) via synthetic data augmentation that transforms SAE text into AAL text.

The reviewers bring up a number of concerns around presentation and clarity, that I agree with -- the graphs could be easier to parse, arguably some of the tables should be graphs, e.g. Fig 1 is a table despite being called a figure and in all cases it should be more clearly indicated what the metrics are and whether higher or lower is better.
There are insufficient methodological details regarding fine tuning, which harms transparency and replicability, please add those at least to the appendix.

However, these are quite minor changes (though important) and can easily be done before publication in another revision, which I recommend that the authors do. none of these complaints are about soundness of scientific rigor.

Given that the work is sound, I believe there is real value in this approach to the research community, which has struggled with what to do about dialects beyond SAE for some time and has really done very little about it. Synthetic data augmentation is a commonly used approach in LLM post training, and this is an interesting linguistically motivated one that is impelementable and works quite well at addressing a real problem.

The only remaining concerns about publication from the reviews are wanting them to have used models larger than Mistral and Chat GPT, which is frankly a ridiculous complaint, everyone does experiments on smaller models and they are of sufficient size to generalise. And a few concerns about validity of using sentiment with no ground truth labels (which the authors address in a rebuttal) and the engagement with the AAL community, which they also address. So I am discounting these concerns as solved.